# Cross-National Associations between Age at Marriage and Intimate Partner Violence among Young Women: An Analysis of Demographic and Health Surveys from 48 Countries

**DOI:** 10.3390/ijerph20043218

**Published:** 2023-02-12

**Authors:** Carolina V. N. Coll, Andrea Wendt, Thiago M. Santos, Amiya Bhatia, Aluisio J. D. Barros

**Affiliations:** 1International Center for Equity in Health, Federal University of Pelotas, Pelotas 96010-610, RS, Brazil; 2Postgraduate Program in Epidemiology, Federal University of Pelotas, Pelotas 96010-610, RS, Brazil; 3Programa de Pós-Graduação em Tecnologia em Saúde (PPGTS), Pontifícia Universidade Católica do Paraná, Curitiba, 80215-901, PR, Brazil; 4Department of Global Health and Development, London School of Hygiene and Tropical Medicine, 15-17 Tavistock Place, London WC1H 9SH, UK

**Keywords:** gender-based violence, early marriage, intimate partner violence, violence against women, low and middle-income countries, gender inequalities

## Abstract

We estimated the associations between age at first marriage and recent intimate partner violence (IPV) among women young women aged 20–24 years using data from demographic and health surveys (DHS) conducted at 48 low- and middle-income countries (LMICs). We fitted a multilevel logistic regression model controlling for sociodemographic covariates. Our pooled analyses revealed that age at marriage is strongly associated with past year IPV in a non-linear way, with steep reductions in violence when young women marry after age 15 and a continued decline in IPV for every year marriage is delayed up to age 24. The risk of physical IPV was 3.3 times higher among women married at age 15 (24.4%, 95% CI 19.7; 29.2%) compared to young women married at age 24 (7.5%, 95% CI 5.8; 9.2%). For sexual IPV, girls married at 15 had 2.2 times higher risk compared to those married at 24 (7.5%, 95% CI 5.6; 9.5% vs. 3.4%, 95% CI 2.7; 4.2%, respectively). For psychological IPV, the relative risk was 3.4 for the same comparison (married at 15: 20.1%, 95% CI 14.6; married at 24: 25.5% vs. 6.0%, 95% CI 3.4; 8.6%). Country specific analyses showed that, age at marriage was negatively associated with physical and psychological IPV in nearly half of the countries (n/48) and with sexual IPV in ten countries. Our findings underscore the importance of integrating violence prevention and response interventions into efforts to prevent child marriage, as well as the health, educations and social services young women access.

## 1. Introduction

Globally, 1 in 3 women, around 736 million, experience physical or sexual violence by an intimate partner during their lifetimes, and intimate partner violence (IPV) is the most prevalent form of violence against women [1]. Adolescence is a period of increased vulnerability to violence, [2] and in most countries, younger women are at higher risk of violence [3]. Recent global estimates of violence against women compiled by WHO show that 1 in 4 young women (aged 15–24 years) who have been in a relationship will already have experienced IPV by the time they reach their mid-twenties, with women in low- and middle-income countries (LMICs) being more affected than those in richer countries [1].

Gender inequality, patriarchal structures, and restrictive gender norms can exacerbate the risk of different forms of gender-based violence (GBV), including child marriage, which is defined as marriage before the age of 18 years and is considered a violation of girls’rights [4]. Child marriage is declining globally but is still common in many countries and has been linked to a higher risk of exploitation, abuse, and violence [5,6,7,8]. Evidence also suggests that child marriage is closely related to the end of formal education and early motherhood and has negative intergenerational effects adversely impacting the socio-economic well-being of women and their children [8,9,10,11,12]. Girls who marry or are in a union before age 18 may be at increased risk of IPV through several potential pathways [8]. First, large power imbalances within relationships and limited women’s agency and autonomy, linked to patriarchal systems and structures, have been found to increase the risk of IPV [13]. Secondly, large age and education gaps between girls and their husbands are common in child marriages, which can further fuel power imbalances within relationships and lead to an increased risk of IPV (since husbands are usually older and more educated) [8]. Thirdly, in contexts where child marriage is common, individuals, families, and communities may hold views and beliefs that violence against the wife is justified in several situations, also increasing the risk of experiencing IPV [14,15].

Despite most studies conducted in individual countries pointing to positive associations between child marriage and IPV. In references [16,17,18], the varying methodological approaches used (e.g., IPV and child marriage measures and definitions) make the reported associations challenging to compare or assess across countries and regions. More recently, a few multi-country studies using comparable data emerged on this topic. In 2017, Kidman [8] used nationally representative samples from 34 countries across all world regions and found that girls married before the age of 18 from most of the study countries were more likely to report physical and/or sexual IPV in the previous year [8]. However, there was considerable heterogeneity in the reported associations between countries; this was particularly evident in sub-Saharan Africa, which was further confirmed in a recent analysis of countries from the region [19]. Associations with psychological IPV were not investigated in this study. More recently, Hayes and Proto [15] conducted a pooled multilevel analysis of national survey data from LMICs to investigate predictors of lifetime IPV and found that a later age of marriage (>18) was linked to a decreased likelihood of experiencing physical and emotional abuse even after controlling for community-level factors and national-level norms and policies on child marriage [15]. Country-specific effects were not investigated by the authors along with the pooled analysis.

The current study conducted secondary data analysis from 48 LMICs to estimate the associations of early marriage and the experience of recent physical, sexual, and psychological IPV among young women aged 20–24 years. This age range is in accordance with the SDG targets 5.2 and 5.3 and United Nations measurement goals to track progress on ending child marriage by 2030 [20]. We add to the existing literature by investigating the age at first marriage in a continuous form (instead of using a cutoff point for child marriage), allowing for greater flexibility to model the relationship between IPV and age at first marriage and for the opportunity to include data from recently conducted surveys, providing a recent panorama of the evidence covering all world regions. Finally, we used a multilevel approach to consider variations between countries while also estimate a pooled result that indicates overall trends based on data from 48 countries.

## 2. Data and Methods

We used data from Demographic and Health Surveys (DHS, http://dhsprogram.com; accessed on 25 January 2023) carried out in LMICs from 2010 onwards. We included all surveys with data on child marriage and domestic violence. For countries with more than one survey, the most recent was selected. DHS surveys collect comparable data across countries based on standard questionnaires. The collection of data on domestic violence has followed a standard module and methodology since 2000 [21]. Given the sensitivity of the questions, one randomly selected ever-married or cohabiting woman per household is selected to participate. Our study population, therefore, includes ever-married and cohabiting young women aged 20–24 years who were: (a) usual residents of the sampled households, (b) eligible for individual interviews with the full women’s questionnaire, (c) were randomly selected for the domestic violence module. Women were interviewed in a private place with no other person present, following the WHO guidelines for the conduct of IPV research.

We estimated the prevalence of past year IPV among women aged 20–24 years as the proportion of ever-married women who reported at least one act of violence from a current or former partner in the 12 months preceding the survey. Women were asked about experiences of recent emotional, physical, and sexual IPV. Physical IPV included experiencing any of seven acts: being pushed, shoved, or having something thrown at her; being slapped; having the arm twisted or the hair pulled; being hit with a fist or something else that could hurt; being kicked, dragged, or beaten up; being choked or burnt on purpose; or being threatened or attacked with a gun, knife, or another weapon. Sexual IPV included being physically forced to have sexual intercourse when not wanted to; having sexual intercourse through coercion or out of fear of what the partner might do, and/or being physically forced to perform any other sexual acts considered humiliating or degrading. Psychological IPV included being humiliated in from of others; being threatened with having someone you care about hurt; or being insulted or made to feel bad about yourself. The experience of these three forms of past year IPV was evaluated separately.

Our main exposure, age at first marriage or union, was recorded in complete years and included as a continuous variable in the analytical models.

We controlled for potential confounders including area of residence (rural or urban, as defined by each country), woman’s age (in years), woman’s education (whether they completed primary school or not, given the age range of the women in study), and socioeconomic position: based on asset indices, obtained from information on household appliances, characteristics of the building materials, presence of electricity, water supply, and sanitary facilities, among other variables [22,23]. Because relevant assets may vary in urban and rural households, separate principal component analyses were carried out in each area, which were later combined into a single score using a scaling procedure to allow comparability between urban and rural households. This score was then used to divide the households into quintiles, weighting by the household size [24].

We estimated the mean age at first marriage and the prevalence of each type of IPV for each country. To explore the relationship between IPV and age at marriage, we fitted a multilevel logistic regression model, controlling for covariates. This pooled model was obtained using a dataset with all countries combined, and sampling weights were adjusted to take into account the population size of women aged 20–24 years in each country in 2015 (the surveys’ median year). Population estimates were obtained from the World Bank database [25]. The multilevel models had women as the first level and the countries as the second level. Given the relationship between IPV levels and age at first marriage could be non-linear, we used fractional polynomials to determine the best parametrization for this variable (see Appendix A for more information). For the model with the best fit, we calculated the adjusted IPV prevalence for each age of marriage between 10 and 24 years and respective confidence intervals. We plotted the results in a graph where key age points (15, 18, 20, and 24 years old) were highlighted in the curve.

We also fitted a logistic regression model with fractional polynomials, controlling for covariates in each country. We calculated the country specific average adjusted IPV prevalence for young women first married at 15 years of age and 24 years of age. The difference between these proportions was calculated, allowing us to estimate a prevalence difference to compare the prevalence of violence among women married at 15 versus 24 years in each country.

All models were fitted using individual level data with the full sample available in each country, therefore increasing statistical power. Analysis used the domestic violence weights and accounted for the complex sample design. Analyses were carried out in Stata 16.1 (StataCorp, College Station, TX, USA) using the *fp* prefix to fit fractional polynomial models and the *margins* command for estimation of predicted values and their confidence intervals. Graphs were created in R version 4.1.0 (R Foundation for Statistical Computing, Vienna, Austria).

Ethical clearance was obtained by national agencies responsible for conducting each survey. The data used for the analysis are publicly available and anonymized to protect participants’ confidentiality.

### 2.1. Role of Funding Source

The funder of the study had no role in study design, data collection, data interpretation, or writing of the report. The corresponding author had full access to all the data in the study and had final responsibility for the decision to submit for publication.

### 2.2. Patient and Public Involvement Statement

Patients or the public were not involved in the design, implementation, or dissemination of this study.

## 3. Results

We included data from 56,171 married women aged 20 to 24 years from surveys in 48 LMICs. The mean age at first marriage and the prevalence of each type of IPV are presented in Table 1 for each country.

The lowest mean age at first marriage among women aged 20–24 years was observed for Chad (15.7; SD 2.3), followed by Mali (16.7; SD 2.4) and Nigeria (16.7; SD 2.8), all in West and Central Africa. The highest mean age at first marriage (19.8; SD 1.5) was in the Maldives in South Asia. In 28 out of 48 countries, the mean age of marriage among women aged 20–24 years was below 18 years of age. The proportion of women aged 20–24 years who experienced IPV in the past year varied widely across countries, including among countries in the same world region. Physical IPV ranged from 2.4% in Armenia to 51.8% in Liberia. In 13 countries more than 25% of the women reported physical IPV in the past 12 months. Psychological IPV ranged from 4.5% in Armenia to 47.8% in Papua New Guinea. More than 25% of women reported psychological IPV in the past year in 15 out of the 48 countries. Sexual IPV ranged from 0.3% in Armenia to 21.7% in Papua New Guinea. Despite the lower mean prevalence in comparison to the other two types of IPV, we found very high prevalence in three countries (Congo DR, Burundi, and Papua New Guinea) with one in five women reporting sexual IPV.

Figure 1 presents the results of the pooled adjusted multilevel logistic regression on the relationship between each type of IPV and age at first marriage using the combined individual-level data from all countries. The figure shows how the prevalence of violence changes with age at marriage. For all three types of violence, past year IPV decreased as the age at first marriage increased. The relative risk of past year physical IPV was 3.3 times higher among women married at age 15 (24.4%, 95% CI 19.7; 29.2%) compared to women married at age 24 years (7.5%, 95% CI 5.8; 9.2%). For past year sexual IPV, women married at 15 years had a risk 2.2 times higher compared to those married at 24 (7.5%, 95% CI 5.6; 9.5% vs. 3.4%, 95% CI 2.7; 4.2%, respectively). For past year psychological IPV, the relative risk observed was 3.4 times higher (married at 15 years: 20.1%, 95% CI 14.6; married at 24 years: 25.5% vs. 6.0%, 95% CI 3.4; 8.6%).

Country-specific adjusted prevalence differences (IPV for women married at 15–IPV for women married at 24) are presented in Figure 2, Figure 3 and Figure 4. In 43 out of 48 countries, the prevalence of physical IPV was higher among women married at 15 years in comparison to those married at 24 years, and in 20 countries these positive associations were statistically significant. The pooled estimate indicated an overall prevalence gap of 16.9 percentage points (95% CI 13.3–20.5) comparing women married at 15 to those married at age 24. Marriage at 15 years compared to 24 years was associated with an increase in the risk of sexual IPV in 34 of the countries studied with statistically significant prevalence differences found in 11 countries. The pooled prevalence difference for all countries was 5.9 (95% CI 3.4–8.4). For psychological IPV, an increased risk of IPV was found in 40 out of 48 countries with significant associations observed in 20 of them. The pooled prevalence gap was 10.6 percentage points comparing women married at 15 years to those married at age 24 years.

Appendix A shows country-specific prevalence differences and confidence intervals and the absolute number of women who experienced each type of violence.

## 4. Discussion

We drew on data from 56,171 married women aged 20 to 24 years from surveys in 48 countries and found that, overall, past year IPV across all three domains (physical, sexual, and psychological) decreases as age at first marriage increases, with more pronounced differences when the age of marriage is between 20 and 24 years. The associations between early marriage and increased IPV remained statistically significant even after adjusting for sociodemographic covariates (wealth, education levels, area of residence, and current age of women). Our findings offer clear evidence that levels of IPV are at their highest for women married younger than age 15. At a first glance, the curves of physical and sexual IPV may suggest that IPV prevalence is lower for women married younger than 15 years (reaching a peak at age 15). However, the confidence intervals indicate that such an interpretation is not correct. In fact, due to the small sample size, it is not possible to infer any difference in IPV prevalence between the ages of 10 and 15 years, only that it is at its highest compared to the other age groups. Young women who are married later, before age 24, have a lower risk of violence compared to those married before 15. However, even among women who were not married at 15, but married before age 24, the risk of violence is still high, ranging from a prevalence of past year IPV from 3.4% to 7.5% for the three types of violence.

The country-specific analysis comparing women married at 15 to those married at 24 showed that, in nearly half of the countries included in the study, child marriage was significantly associated with increased past year physical and psychological IPV among young women and in 10 out of the 48 countries presented positive associations for sexual IPV.

Many of the countries with significant associations between early marriage and IPV were in Latin America, the Caribbean, and South Asia. In the South Asia region, there were significant prevalence gaps (more violence reported among girls married at 15 in comparison to those married at 24) in India and Afghanistan for all the three domains of IPV, while a significant association between early marriage and past year psychological IPV was observed in Pakistan. All six countries from Latin America and the Caribbean included in our analysis presented a significant positive association for at least one of the IPV domains. In Colombia, an increased risk of IPV was observed for the three IPV domains. There were also several countries from West and Central Africa, as well as from Eastern and Southern Africa, for which consistent positive associations were observed across all domains of violence. Zambia, Malawi, Nigeria, and Tanzania showed positive associations for physical, sexual, and psychological IPV. Although a higher heterogeneity was observed for the other world regions, early marriage was linked to increased risks of IPV in a considerable number of countries. It is also important to note that for those countries in which significant associations were not found, the risk ratios pointed to null effects. Associations in the opposite direction—child marriage decreasing the risk of IPV—were not found in our study.

Our results add to the previous evidence on the negative impacts of child marriage on young women globally [8,15,26,27,28]. Worldwide, efforts to address child marriage include interventions to prevent marriage before age 15 or age 18, and to delay the age of marriage. These have included skill building (education, life skills, vocational training), asset-building, conditional and unconditional cash transfers, advocacy with parents and community leaders, social norm change, social welfare schemes, youth-friendly sexual and reproductive health services, school vouchers, and girls/youth clubs [29,30]. However, heterogeneous impacts depending on the social context have been found and need to be considered [31]. Our findings underscore the importance of integrating violence prevention and response interventions into efforts to prevent and delay child marriage, as well as the health, education, and social services that young women access. Given the increased risk of IPV that young married women encounter, ensuring that violence services are youth-friendly is also crucial. Focusing solely on efforts to prevent and delay child marriage to prevent violence among married women will not be enough, as the prevalence of IPV remained high even among women who married at 24. More importantly, our findings point to the importance of violence prevention interventions across the life course both within and outside the household. Such interventions should also be youth-friendly and consider the contexts of girls who are married before 18 years. For example, a study in Cote d’Ivoire found that an intervention to prevent IPV was less effective among girls who had an early marriage [32], emphasizing the reciprocal need to consider early marriage in the design and delivery of IPV prevention interventions.

Some limitations should be highlighted for an accurate interpretation of the findings. First, due to the cross-sectional nature of the study, causality should not be inferred for the associations investigated. However, this limitation was minimized by limiting our sample to women of at least 20 years and ensuring that child marriage would precede recent IPV exposure, thereby respecting exposure–outcome temporality. To account for the possibility of bias introduced by some women who had been in a union for less than 12 months (the recall period for the IPV assessment), we also performed a sensitivity analysis that excluded women for whom the difference between the current age and the age of first marriage was less than 12 months. The findings were virtually the same (Appendix A). Second, given the challenges involved in the reporting of violence, it is possible that the prevalence of IPV is underestimated and, if women married as a child are less likely to disclose, a weakening in the associations is expected. The IPV risk differences (marriage at 15 years–marriage at 24 years) that we estimated are based on country-specific curves, and in several of them, the number of women who experienced IPV is very small, particularly for sexual IPV. In Maldives, for example, only four women reported sexual IPV, resulting in a 0.3 prevalence at 15 years, which may explain the null effects found. We presented this information in the Supplementary Material and highlight that although this country-specific estimates are imprecise, the pooled curve is based on a much larger sample with more accurate estimates. Third, given its high correlation to early marriage, marriage duration might also be a potential source of bias while studying young cohorts of women. To account for the possibility of child marriage acting as a proxy for years since marriage, we repeated the pooled analysis for an older cohort of women (25–39 years). Although findings suggest weaker associations in comparison to younger women, associations are still significant for physical IPV and are borderline for sexual IPV (Appendix A). Finally, we should also be aware that these are adjusted estimates that apply for married women aged 20–24 according to their age of marriage and are not estimates for the levels of intimate partner violence experienced by women in general.

However, we use data from DHS surveys with a standardized methodology, allowing the inclusion of many countries with comparability between them. Further, as far as we know, this is the first study to explore the age at first marriage as a continuous variable and that presents both country-specific and pooled findings for 48 LMICs. Upon using this approach, it became evident that the likelihood of a girl experiencing IPV is delayed for each increase in the age at first marriage. This approach diverges from most studies in the literature, which rely on specific cut points to analyze the potential consequences of child marriage (e.g., ages 15 or 18) [15]. In comparison to the previous multi-country studies, our findings also provide more robust evidence of the significant associations for all types of IPV (physical, sexual, and emotional), which were investigated separately.

## 5. Conclusions

Our study contributes to reinforcing the links between early marriage and IPV and underscores the importance of violence prevention interventions focused on young women. By addressing the intersections between girl-child marriage and IPV victimization, our findings also contribute to greater coordination between fields of violence against children and violence against women.

## Figures and Tables

**Figure 1 ijerph-20-03218-f001:**
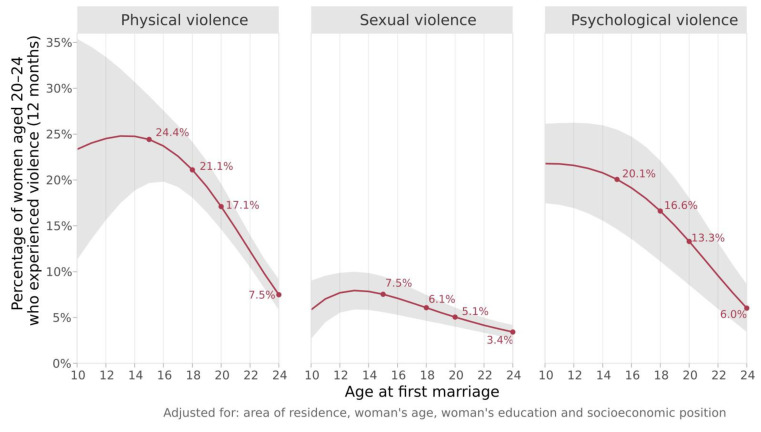
Average adjusted past year IPV prevalence according to age at first marriage for women aged 20 to 24 years. Pooled results for 48 low- and middle-income countries obtained from a multilevel logistic regression model. The red line represents the continuous IPV estimate from ages 10 to 24 years at first marriage. The grey area represents the respective confidence interval. IPV estimates for the ages 15, 18, 20, and 24 years are highlighted.

**Figure 2 ijerph-20-03218-f002:**
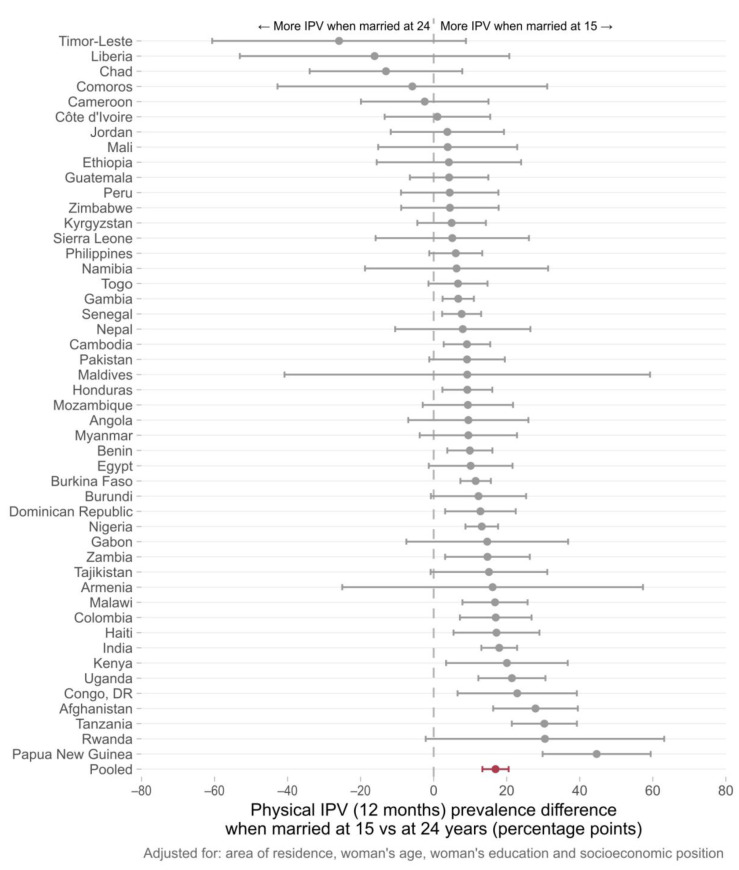
Difference between the average physical IPV prevalence among women aged 20–24 years first married at 15 years and at 24 years of age. Individual results for 48 low- and middle-income countries obtained from logistic regression models. The pooled estimate of the prevalence difference in past year physical IPV (in red) was 16.9 percentage points (95% CI 13.3–120.5).

**Figure 3 ijerph-20-03218-f003:**
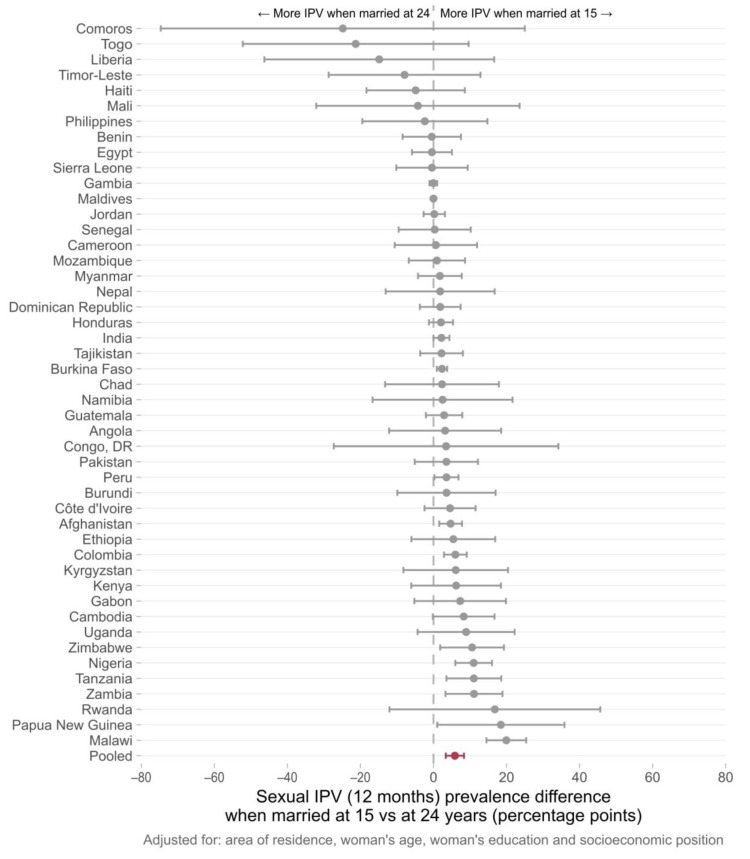
Difference between the average sexual IPV prevalence among women aged 20–24 years first married at 15 years and at 24 years of age. Individual results for 48 low- and middle-income countries obtained from logistic regression models. The pooled estimate of the prevalence difference in past year sexual IPV (in red) was 5.9 percentage points (95% CI 3.4–8.4).

**Figure 4 ijerph-20-03218-f004:**
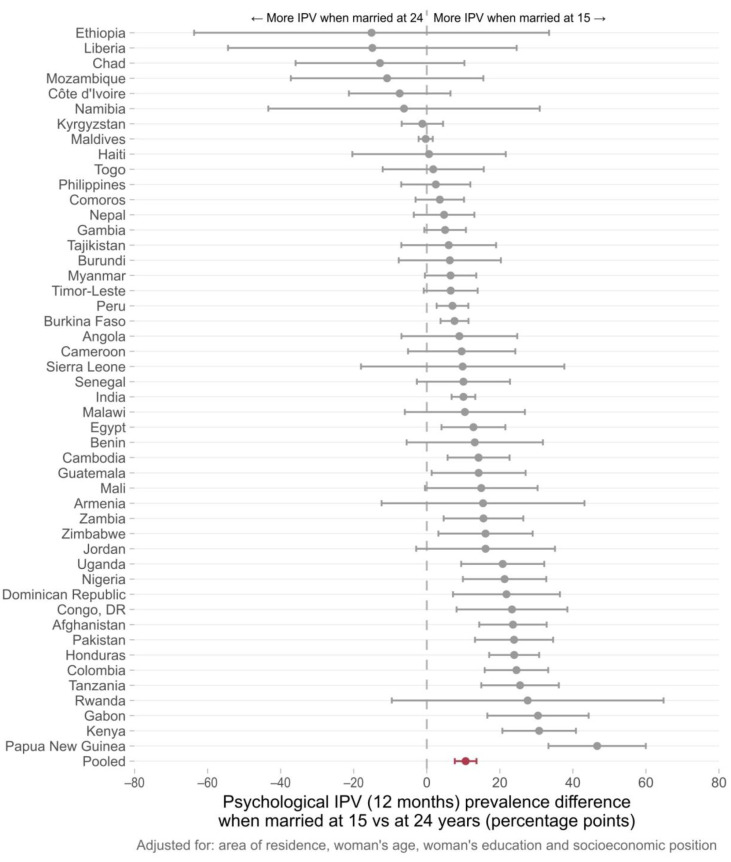
Difference between the average adjusted psychological IPV prevalence among women aged 20–24 years first married at 15 years and at 24 years of age. Individual results for 48 low- and middle-income countries obtained from logistic regression models. The pooled estimate of the prevalence difference in past year psychological IPV (in red) was 10.6 percentage points (95% CI 7.7–13.6).

**Table 1 ijerph-20-03218-t001:** Child marriage and intimate partner violence among women aged 20–24 years by UNICEF world regions. Demographic and Health Surveys 2010–2018.

		Age at First Marriage		Past Year IPV			
	Year	Mean	(sd)	Physical	Sexual	Psychological	N
**West & Central Africa**							
Benin	2017	17.3	3.0	9.2	6.3	27.1	717
Burkina Faso	2010	17.0	2.2	9.6	1.9	7.2	2074
Cameroon	2018	17.0	2.8	20.8	6.8	21.4	738
Chad	2014	15.7	2.3	16.7	7.9	18.4	703
Congo, DR	2013	17.0	2.4	37.4	20.5	34.9	1093
Côte D’Ivoire	2011	17.1	2.7	25.4	6.9	15.9	871
Gabon	2012	18.0	2.8	38.3	11.9	27.3	649
Gambia	2013	17.5	2.8	6.0	1.1	7.1	654
Liberia	2019	17.6	2.5	51.8	11.3	45.1	349
Mali	2018	16.7	2.4	21.4	9.3	26.7	620
Nigeria	2018	16.7	2.8	15.3	8.4	29.2	1306
Senegal	2019	17.5	2.7	5.1	3.6	7.8	214
Sierra Leone	2019	17.2	2.8	46.8	7.0	47.0	589
Togo	2013	18.0	2.6	12.1	5.9	22.2	769
**Eastern & Southern Africa**							
Angola	2015	17.5	2.7	30.8	10.1	27.3	1613
Burundi	2016	18.1	2.2	23.3	21.5	18.8	1197
Comoros	2012	17.1	2.8	6.1	2.3	7.3	429
Ethiopia	2016	16.8	2.8	20.1	7.3	20.5	839
Kenya	2014	18.2	2.5	23.9	10.5	23.6	739
Malawi	2015	17.3	2.4	16.8	16.5	21.2	1212
Mozambique	2011	17.1	2.7	28.8	8.4	34.0	1101
Namibia	2013	18.8	2.7	23.3	4.6	20.7	170
Rwanda	2014	19.6	2.0	18.2	7.8	18.1	274
Tanzania	2015	17.8	2.3	31.2	12.3	29.7	1354
Uganda	2016	17.7	2.5	25.4	18.6	31.0	1617
Zambia	2018	17.7	2.4	24.4	12.6	23.6	1396
Zimbabwe	2015	17.8	2.2	20.3	10.4	25.0	995
**Middle East & North Africa**							
Egypt	2014	18.5	2.3	20.1	3.1	14.9	924
Jordan	2017	18.8	2.5	10.7	2.1	16.1	715
**Eastern Europe & Central Asia**							
Armenia	2015	19.5	1.8	2.4	0.3	4.5	316
Kyrgyzstan	2012	19.4	1.7	11.7	1.8	6.9	747
Tajikistan	2017	19.0	1.5	18.2	1.3	11.2	801
South Asia							
Afghanistan	2015	17.3	2.5	40.9	5.6	30.2	3923
India	2015	18.2	2.4	22.3	5.0	10.8	8847
Maldives	2016	19.8	1.5	9.4	1.0	7.1	333
Nepal	2016	17.6	2.4	10.3	5.8	6.0	632
Pakistan	2017	18.4	2.5	10.2	3.0	17.6	555
**East Asia & Pacific**							
Cambodia	2014	18.6	2.4	7.3	3.9	12.0	456
Myanmar	2015	18.4	2.2	12.0	3.6	10.9	340
Papua New Guinea	2016	17.7	3.0	49.0	21.7	47.8	587
Philippines	2017	18.3	2.4	9.7	2.8	9.6	1457
Timor Leste	2016	18.7	2.3	31.6	5.3	8.0	448
**Latin America & Caribbean**							
Colombia	2015	17.9	2.5	32.8	4.5	29.7	3263
Dominican Republic	2013	17.1	2.7	15.5	3.8	29.5	930
Guatemala	2014	17.6	2.5	10.7	2.7	17.1	997
Haiti	2016	18.2	2.5	14.3	8.1	20.2	532
Honduras	2011	17.4	2.5	12.6	2.9	23.2	2085
Peru	2019	18.2	2.4	13.8	2.7	10.3	3001

## Data Availability

All data relevant to the study are included in the article or available as Appendix A. The data used in the analyses are publicly available, anonymized, and geographically scrambled to ensure confidentiality. More information on DHS can be found at “https://dhsprogram.com/ (accessed on 25 January 2023)”, where survey datasets can be obtained.

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
