# Peer review of "Cross-National Associations between Age at Marriage and Intimate Partner Violence among Young Women: An Analysis of Demographic and Health Surveys from 48 Countries"

_ijerph, 2023, doi:10.3390/ijerph20043218_

Round 1
Reviewer 1 Report
The authors have created a unique and interesting study analyzing age of marriage and Intimate Partner Violence in 48 low- and middle-income countries. The authors identify the different types of IPV (psychological, sexual and physical) and have rank-ordered the occurrence.
The graphs clearly demonstrate their findings. Overall, this is an important study indicating the negative influence child marriage and the resulting varying kinds of IPV.
I would suggest that the authors do one more proofread. For example, on page 10, last paragraph, line 271 the sentence reads, “More importantly, our findings point to the important of violence prevention interventions…” This should be corrected to importance.
Despite the topic, this paper was interesting and informative shedding new light on a worldwide problem that continues.
Author Response
Response 1: Thank you to reviewer 1 for the compliments to our work. As suggested, the manuscript was proofread. The sentence on line 271 specified was corrected.
Reviewer 2 Report
The authors present a well designed study with is timely and informative. Very well written manuscript. The purpose and methodology is clear. The tables and figures are easy to follow and support the findings and discussion. Line 240 "data not shown" - explain rationale of perhaps include in supplemental data.
Just a couple of edits needed:
Line 38 - ...18 years, is much more common...
Line 98 - ...in front of others?
Line 259 ...marriage for young women..?
Line 283...by some women who have been in a union...?
Line 294...information in the Appendix and...
Line 318...underscores...
Author Response
The authors present a well designed study with is timely and informative. Very well written manuscript. The purpose and methodology is clear. The tables and figures are easy to follow and support the findings and discussion. Line 240 "data not shown" - explain rationale of perhaps include in supplemental data.
Response 1: Thank you for seeing the valuable contribution of this work and compliments. As suggested, we decided to include the graphs with relative risks in the supplementary material.
Just a couple of edits needed:
Line 38 - ...18 years, is much more common...
Response 2: The sentence was corrected.
Line 98 - ...in front of others?
Response 3: This is the way que question is posed in the instrument.
Line 259 ...marriage for young women..?
Response 4: The sentence was corrected.
Line 283...by some women who have been in a union...?
Response 5: The sentence was corrected.
Line 294...information in the Appendix and...
Response 6: The sentence was corrected.
Line 318...underscores...
Response 7: The word was corrected.
Reviewer 3 Report
This study aimed to estimate the association between age at first marriage and last intimate partner violence (IPV) among young women aged 20-24 years. I think that this topic is so important and represents an important field of study, but I have some comments that I think will contribute to strengthening this paper:
Literature review:
- I noted the weakness of the previous studies section, especially since the paper covers 48 countries in the world, moreover, I did not read enough previous studies that reflect the need for this study in these countries, or at least some of them.
Methodology
It needs to explain the type of method used, and whether data was collected through questionnaires or secondary data. If data was collected through the survey, how were these tools developed, and what is the level of validity and reliability?
Discussion
This section is also weak, due to the weakness of previous studies, which made it difficult for researchers to link the findings of the current study with the findings of previous studies.
Theoretical direction
The current study needs a good theoretical direction that defines the concepts of the study and the theoretical interpretation of its results by adopting some international models in the analysis or building a model through which the results are analyzed.
References
The references are old, most of them before 2019. The references need to be updated
Author Response
This study aimed to estimate the association between age at first marriage and last intimate partner violence (IPV) among young women aged 20-24 years. I think that this topic is so important and represents an important field of study, but I have some comments that I think will contribute to strengthening this paper:
Response 1: Thank you for seeing the valuable contribution of this work.
Literature review:
- I noted the weakness of the previous studies section, especially since the paper covers 48 countries in the world, moreover, I did not read enough previous studies that reflect the need for this study in these countries, or at least some of them.
Response 2: There are a few previous studies carried out in individual countries, mostly from Asia and Africa (re. Yount et al in Bangladesh; Raj et al in India, Tenkorang in Ghana cited in the introduction), suggesting that child marriage is a strong risk factor for IPV. However, there is considerably diversity in their methodological approaches (e.g. IPV and child marriage measurement and definitions, analytical technique) making it difficult to compare the reported associations. In the last years, two cross-country studies using data from nationally representative surveys with standardized definitions and measures of violence emerged and provide country-specific findings (Kidman 2017 and Ahinkorah et al 2021). Although these studies filled the gap of having cross-national comparable data, some of their limitations include the fact that the work of Kidman, did not studied psychological violence and include surveys conducted more than a decade ago (data up till 2013) and therefore do not provide a current picture of the associations. Also, the pooled estimates presented did not considered the variations between countries. On the other hand, while Akinkorah et al use data from more recent surveys, the authors only covered countries from the sub-Saharan Africa region. In addition to this, both studies used a cut-point to define child marriage. We believe the main differential of our work is: 1) to provide both country-specific and pooled estimates of the association between early marriage and IPV; 2) the use of data from recently conducted surveys (all from 2010 onwards) providing a recent panorama of associations covering all regions of the world and the inclusion of data from 14 countries contemplated in these two studies; 3) the use of age at marriage in a continuous format, allowing greater flexibility to shape the relationship between age at first marriage and IPV.
We expanded the introduction section to make comparisons with previous studies and advances being made clearer (see lines 53-82).
Despite most studies conducted in individual countries pointing to positive associations between child marriage and IPV.16–18 the varying methodological approaches used (e.g. IPV and child marriage measures and definitions), make the reported associations challenging to compare or assess across countries and regions. More recently, a few multi-country studies using comparable data emerged on this topic. In 2017, Kidman8 used nationally representative samples from 34 countries across all world regions and found that girls married before the age of 18 from most of the study countries were more likely to report physical and/or sexual IPV in the previous year.8 However, there was considerable heterogeneity in the reported associations between countries, particularly evident in sub-Saharan Africa, which was further confirmed in a recent analysis of countries from the region.19 Associations with psychological IPV were not investigated in this study. More recently, Hayes and Proto15 conducted a pooled multilevel analysis of national survey data from LMICs to investigate predictors of lifetime IPV and found that a later age of marriage (>18) was linked to a decreased likelihood of experiencing physical and emotional abuse even after controlling for community-level factors and national-level norms and policies on child marriage.15 Country-specific effects were not investigated by the authors along with the pooled analysis.
The current study conducts secondary data analysis from 48 LMICs to estimate the associations of early marriage and the experience of recent physical, sexual, and psychological IPV among young women aged 20-24 years. This age range is in accordance with the SDG targets 5.2 and 5.3, and United Nations measurement goals to track progress on ending child marriage by 2030.20 We add to the existing literature by investigating the age at first marriage in a continuous form (instead of using a cutoff point for child marriage), allowing for greater flexibility to model the relationship between IPV and age at first marriage and for the opportunity to include data from recently conducted surveys, providing a recent panorama of the evidence covering all world regions. Finally, we used a multilevel approach to consider variations between countries while also creating a pooled result that indicates overall trends.
Methodology
It needs to explain the type of method used, and whether data was collected through questionnaires or secondary data. If data was collected through the survey, how were these tools developed, and what is the level of validity and reliability?
Response 3: Thank you for your suggestion. We now specified that we used secondary data collected from Demographic and Health Surveys and provided more details on the methods used for data collection (lines 72; 89-90).
Discussion
This section is also weak, due to the weakness of previous studies, which made it difficult for researchers to link the findings of the current study with the findings of previous studies.
Response 4: Thank you for pointing this limitation. Given the mentioned challenges in comparing data from different studies (see response 2) we opted to focus the discussion on comparing the findings from the different countries and regions. However, the introduction was now expanded to provide more details of previous studies as well as advances of the current study.
Theoretical direction
The current study needs a good theoretical direction that defines the concepts of the study and the theoretical interpretation of its results by adopting some international models in the analysis or building a model through which the results are analyzed.
Response 5:The association between early marriage and increased IPV is well established in the literature, and our introduction was revised to make this even more clear. Our study advances the field mainly by looking at age as a continuous variable instead of using it dichotomized as most other studies did (e.g. marriage before age 15), and important lessons came out of this approach. Because this is a single predictor model, we do not see the need for a conceptual model to run the analysis. On the other hand, potential mechanisms discussed in the literature are presented in the introduction (lines 44-52).
References
The references are old, most of them before 2019. The references need to be updated
Response 6: Even though most of the studies investigating the associations between child marriage and IPV were not recently published we believe our literature review is covering the most relevant studies in the field. In fact, two of the three cross-country studies cited were conducted after 2019 (Akinkorah and Hayes & Protas). In any case, we conducted an additional search for recently published studies and, although not specifically investigating the association of interest, we added some studies bringing the most updated evidence on child marriage which were published in a recent supplement of the Journal of Adolescent Health (https://www.jahonline.org/issue/S1054-139X(21)X0013-4):
Psaki SR, Melnikas AJ, Haque E, et al. What Are the Drivers of Child Marriage? A Conceptual Framework to Guide Policies and Programs. J Adolesc Heal 2021; 69. DOI:10.1016/j.jadohealth.2021.09.001.
Malhotra A, Elnakib S. 20 Years of the Evidence Base on What Works to Prevent Child Marriage: A Systematic Review. J Adolesc Heal 2021; 68: 847–62.
Makino M, Ngo TD, Psaki S, Amin S, Austrian K. Heterogeneous Impacts of Interventions Aiming to Delay Girls’ Marriage and Pregnancy Across Girls’ Backgrounds and Social Contexts. J Adolesc Heal 2021; 69. DOI:10.1016/j.jadohealth.2021.09.016.
Round 2
Reviewer 3 Report
No more comments